# Learning a Non-Redundant Collection of Classifiers

## Abstract

Supervised learning models constructed under the i.i.d. assumption have often been shown to exploit spurious or brittle predictive signals instead of more robust ones present in the training data. Inspired by Quality-Diversity algorithms, in this work we train a collection of classifiers to learn distinct solutions to a classification problem, with the goal of learning to exploit a variety of predictive signals present in the training data. We propose an information-theoretic measure of model diversity based on minimizing an estimate of conditional total correlation of final layer representations across models given the label. We consider datasets with synthetically injected spurious correlations and evaluate our framework's ability to rapidly adapt to a change in distribution that destroys the spurious correlation. We compare our method to a variety of baselines under this evaluation protocol, showing that it is competitive with other approaches while being more successful at isolating distinct signals. We also show that our model is competitive with Invariant Risk Minimization (IRM) under this evaluation protocol without requiring access to the environment information required by IRM to discriminate between spurious and robust signals.

## 1 Introduction

The Empirical Risk Minimization (ERM) principle (Vapnik, 2013), which underpins many machine learning models, is built on the assumption that training and testing samples are drawn i.i.d. from some hypothetical distribution. It has been demonstrated that certain violations of this assumption (in conjunction with potential misalignments between the formulation of the learning objectives and the underlying task of interest) lead to models that exploit spurious or brittle correlations in the training data. Examples include learning to exploit image backgrounds instead of objects in the foreground due to data biases (such as using grassy backgrounds to predict the presence of cows (Beery et al., 2018)), using textural as opposed to shape information to classify objects (Geirhos et al., 2018), and using signals not robust to small adversarial perturbations (Ilyas et al., 2019).

Implicit in work that attempts to address these phenomena is the assumption that more robust predictive signals are indeed present in the training data, even if for various reasons current models do not have the tendency to leverage them.

In this work, drawing inspiration from Quality-Diversity algorithms (Pugh et al., 2016) – which seek to construct a collection of high-performing, diverse solutions to a task – we aim to learn a collection of models, each incentivized to find a distinct, high-performing solution to a given supervised learning problem from a fixed training set. Informally, our motivation is that a sufficiently large collection of such distinct models would exploit robust signals present in the training data in addition to the brittle signals that current models tend to exploit. Thus, given the representations computed by such a collection, it may be possible to rapidly adapt to test-time shifts in distribution that destroy the predictive power of brittle features.

Addressing this problem hinges on defining and enforcing an appropriate measure of model diversity. To this end, we make the following contributions:

- We propose and motivate a novel measure of model diversity based on conditional total correlation (across models) of final layer representations given the label. Informally, this

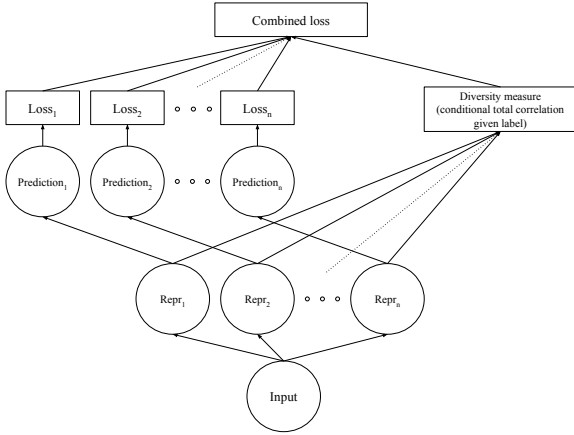

Figure 1: Graphical representation of our framework. We train multiple models to individually minimize supervised loss while simultaneously minimizing conditional total correlation of final layer representations given the label.

  incentivizes models to each learn a non-redundant (in an information-theoretic sense) way to solve the given task.

- We estimate this measure using a proxy variational estimator computed using samples from the conditional joint and marginal distributions of final layer representations across models. We train a collection of models to be accordingly diverse, alternating between training a variational critic to maximize the variational estimator and minimizing a weighted sum of the classification losses across models and the variational estimator.

- We empirically validate this framework by training on datasets with synthetically injected spurious correlations. We demonstrate that our framework is able to learn a collection of representations which, under a linear fine-tuning protocol, are competitive with baselines in being able to rapidly adapt to a shift in distribution which destroys the spurious correlations. We show that our framework performs favourably relative to baselines in being able to isolate distinct predictive signals in different models in the collection, as a result being able to make accurate test-time predictions without fine-tuning. We also compare our approach to the Invariant Risk Minimization (Arjovsky et al., 2019) framework which leverages information from multiple training environments to identify signals robust to variations across environments. We show that our approach is able to exploit the robust signals without requiring the metadata needed by IRM to discriminate between spurious and robust signals.

## 2 TOTAL CORRELATION

Total correlation (Watanabe, 1960) is a multivariate extension of mutual information. The total correlation for $n$ random variables $X_1, ..., X_n$ is given by (where $KL$ denotes the Kullback-Leibler divergence):

$$TC(X_1, ..., X_n) := KL[p(X_1, ..., X_n) \| \prod_{i=1}^{n} p(X_i)] \qquad (1)$$

For $n = 2$ this is equivalent to mutual information. We can interpret this as a measure of the amount of redundancy across the $X_i$'s. A total correlation of zero corresponds to the $X_i$'s being mutually independent.

Given a classification problem predicting label $Y$ from $X$, we consider the conditional total correlation of a collection of vector-valued representations $h_1(X), ..., h_n(X)$ given $Y$ for differentiable functions $h_i$, defined as:

$$TC(h_1(X), ..., h_n(X)|Y) := \mathbb{E}_Y[KL[p(h_1(X), ..., h_n(X)|Y)\|\prod_{i=1}^{n} p(h_i(X)|Y)]] \qquad (2)$$

where $X$ is drawn from the conditional distribution given $Y$.

## 3 METHOD

Consider a collection of differentiable representation function and linear classifier pairs $\{(h_i, c_i)\}_{i \in \{1, ..., n\}}$ where we denote the vector of parameters of all these functions as $\theta \in \mathbb{R}^m$, such that for a classification problem predicting $Y$ from $X$, we compute logits $c_i \circ h_i(X)$. That is, the $h_i$ computes a vector-valued representation of the input $X$, and $c_i$ transforms this into a scalar classification prediction.

We target the following optimization problem:

$$\min_{\theta} \qquad \sum_{i=1}^{n} \mathbb{E}_{X,Y}[l(c_i \circ h_i(X), Y)]$$
$$\text{subject to} \quad TC(h_1(X), ..., h_n(X)|Y) < \epsilon \qquad (3)$$

for some $\epsilon > 0$, where $l$ is some loss function.

That is, we seek to minimize the risk for each model while keeping the conditional total correlation of final layer representations given the label below some threshold.

We justify the choice of conditional total correlation in place of the unconditional version by appealing to the following argument: if we consider the case where the collection consists of just two models, total correlation is equivalent to mutual information. Minimizing mutual information between the two representations is equivalent to enforcing independence. However, it is not possible for the representations to be independent while requiring that both are sufficient to accurately predict the label. On the other hand, provided that the representations are not dependent due to some common cause, they will be conditionally independent given the label. Thus, minimizing the conditional total correlation should provide a more useful learning signal.

In practice, we attempt to solve the following unconstrained problem in place of (3):

$$\min_{\theta} \quad \sum_{i=1}^{n} \mathbb{E}_{X,Y}[l(c_i \circ h_i(X), Y)] + \beta TC(h_1(X), ..., h_n(X)|Y) \qquad (4)$$

where $\beta > 0$ is a hyperparameter and we estimate the risk using the empirical risk (i.e. sample mean) .

### 3.1 TOTAL CORRELATION ESTIMATION

In initial experiments with the two-model case (i.e. where total correlation is equivalent to mutual information), the InfoNCE objective of Oord et al. (2018), shown to compute a lower bound of the mutual information (Poole et al., 2019), proved effective in computing useful gradient estimates for the latter term in (4). Consequently, we choose to construct a proxy estimator for the total correlation based on InfoNCE.

The InfoNCE objective for random variables $U$ and $V$ is given by:

$$I_{\text{NCE}} := \frac{1}{K} \sum_{i=1}^{K} \log \frac{e^{f(u_i, v_i)}}{\frac{1}{K} \sum_{j=1}^{K} e^{f(u_i, v_j)}} \qquad (5)$$

where $(u_i, v_i)$ are sampled from the joint distribution of $U$ and $V$, $K$ is the batch size, and $f$ is a differentiable, scalar-valued function, commonly referred to as a 'critic' (Poole et al., 2019). The parameters of $f$ are optimized to maximize $I_{\text{NCE}}$.

By analogy, our proposed estimator $\widehat{TC}(X_1, ..., X_N)$ is given by:

$$\widehat{TC}(X_1, ..., X_N) := \frac{1}{K} \sum_{i=1}^{K} \log \frac{e^{f(x_{i1}, ..., x_{in})}}{\frac{1}{M} \sum_{j=1}^{M} e^{f(x_{\pi_{1,j}1}, ..., x_{\pi_{n,j}n})}} \tag{6}$$

where $(x_{i1}, ..., x_{in})$ are sampled from the joint distribution of $X_1, ..., X_n$, $K$ is the batch size, $\pi_{k,j} \sim \text{Uniform}(\{1, ...K\})$ is a random index of the batch dimension, $M$ is a hyperparameter, and $f$ is a critic. The permuted $\pi_{k,j}$ indices are intended to yield $(x_{\pi_{1,j}1}, ..., x_{\pi_{n,j}n})$ samples drawn approximately from the marginals, derived from the observations in the batch. Similarly we optimize the parameters of $f$ to maximize $\widehat{TC}(X_1, ..., X_N)$.

We interpret (6) as a ratio of scores between samples from the joint distribution of the $X_i$'s and samples approximately drawn from the marginals, such that a solution to the variational objective resulting in a large ratio corresponds to being able to easily discriminate between samples from the joint and marginals (i.e. a proxy for large total correlation) and conversely when the ratio is small.

We compute the conditional case as follows:

$$\widehat{TC}(X_1, ..., X_N | Y) := \mathbb{E}_Y \left[ \frac{1}{K} \sum_{i=1}^{K} \log \frac{e^{f(x_{i1}, ..., x_{in})}}{\frac{1}{M} \sum_{j=1}^{M} e^{f(x_{\pi_{1,j}1}, ..., x_{\pi_{n,j}n})}} \right] \tag{7}$$

where $(x_{i1}, ..., x_{in})$ are sampled from the conditional joint distribution of $X_1, ..., X_n$ given $Y$. We perform the conditioning step by splitting a batch by label, separately computing the estimator for each group, and finally computing a weighted average. The weights are estimated using the frequencies of each label in the batch.

## 3.2 MINIMAX TRAINING OBJECTIVE

We replace the latter term in (4) with our variational estimator to obtain:

$$\min_{\theta} \sum_{i=1}^{n} \mathbb{E}_{X,Y}[l(c_i \circ h_i(X), Y)] + \beta \max_{\phi} \widehat{TC}(h_1(X), ..., h_n(X) | Y) \tag{8}$$

where $\phi$ are the parameters of the variational critic $f$, and $\beta > 0$ is a hyperparameter.

In practice, we use gradient-based methods to approximately optimize (8) by alternating between optimizing $\phi$ to maximize the variational estimator (giving us a local total correlation estimate), and optimizing $\theta$ to minimize the sum of all terms (for fixed $\phi$).

## 4 EXPERIMENTS

The main hypothesis that we wish to test is that our method yields a collection of models that exploit and isolate distinct predictive signals in the training data. Our strategy for testing this hypothesis is to consider a training distribution with synthetically injected spurious correlations as well as a test distribution in which the spurious correlation has been destroyed. We then measure our collection of models's ability to rapidly adapt to the new distribution given small amounts of data, relative to a number of baselines. The spurious correlations are constructed such that they are easier to exploit than the robust signals.

The central challenge in testing our main hypothesis is in constructing a reliable method for measuring the extent to which distinct predictive signals have been extracted and isolated in the outputs of trained models. Our evaluation protocol which requires access to a small amount of data from a shifted distribution to construct a model selection strategy emerges from a lack of alternatives. It precludes the use of various real-world benchmarks known or hypothesized to exhibit a variety of predictive signals, spurious or otherwise: we simply do not currently have access to appropriate test sets.

Looking to the related problem of domain generalization for alternate evaluation protocols / model selection strategies, we find approaches either similarly based on oracle test sets or using held-out training domains (Gulrajani & Lopez-Paz, 2020). Here, we deliberately consider a problem setting which does not assume access to training data from multiple distinct but related domains, so we cannot use model selection strategies based on the latter. Indeed, the underlying motivation for the problem setting we consider is that perhaps we can get away with very little additional information in order to isolate distinct predictive signals in a training set. A central priority for future work is to explore the possibility of model selection strategies that do not require access to data from a shifted evaluation distribution. In this paper, we focus on empirical evaluation of our approach relative to baselines under our proposed fast adaptation evaluation protocols.

We consider variations on the Colored MNIST (C-MNIST) task of Arjovsky et al. (2019). This task consists of classifying MNIST digits into one of two classes (0-4 or 5-9). However, the true label is flipped with probability 0.25 and an additional binary signal (colour) is constructed from this corrupted label such that it is more strongly correlated with the corrupted label than the digit. As demonstrated in Arjovsky et al. (2019), a model trained with ERM will overwhelmingly tend to exploit the colour information. In Arjovsky et al. (2019), the goal was to demonstrate an ability to learn to exploit the digit information given the task of identifying a stable signal across environments with varying colour-label correlation. To test our framework, we assume no such access to multiple environments, and collapse the two training environments in the C-MNIST benchmark task of Arjovsky et al. (2019) into one training set.[1] The results for this task are collected in Table 1.

We also consider a modified dataset which introduces an obstacle to extracting mutually exclusive representations of digit and colour. Specifically, we use functions of a new binary 'common cause' variable to both rotate the digits and perturb the colour signal, such that minimizing total correlation will depend on an ability to discard the common information. This is to test our framework's ability to still extract the predictive signals in the presence of such an obstacle to minimizing conditional total correlation. We refer to this dataset as Rotated Colored MNIST (RC-MNIST). The results for this task are collected in Table 2.

We consider a third variation in which more than two predictive signals are present in the training data. Specifically, we construct another binary 'colour' signal from the corrupted label and add it to the input such that new signal can be exploited on the training set for a maximal accuracy of 0.75. We refer to this dataset as Two-Colored MNIST (TC-MNIST). The results for this task are collected in Table 3.

We similarly construct the same three benchmarks using the Fashion-MNIST dataset (Xiao et al., 2017).

We consider the following baselines:

- Single ERM classifier
- Single ERM classifier with DeCov (Cogswell et al., 2015) regularization. We use this to penalize large non-diagonal entries of the covariance matrix of the output of the representation function
- Single ERM classifier with Mutual Angles (MA) (Xie et al., 2015) regularization. We use this to penalize the mean angle between weight vectors corresponding to the output units of the representation function, minus the variance of the angles
- Single ERM classifier with a Uniform Eigenvalue Regularizer (UER) (Xie et al., 2017). We use this to encourage the weight matrix computing the output of the representation function to have uniform eigenvalues
- Ensemble of ERM classifiers
- Collection of ERM classifiers trained to maximize pairwise cosine distances between final layer representations (for a given input), adapted from Yu et al. (2011)

---

[1]The two environments are constructed such that the colour signal is computed by flipping the corrupted label with probability 0.1 and 0.2 respectively. Thus, collapsing the two environments into one means that we can at best achieve an accuracy of 0.85 on the training set by exploiting the colour signal, versus the maximal accuracy of 0.75 by exploiting the digit.

We also include results for the following methods for reference (these methods require access to information that the baselines do not require):

- Importance-weighted ERM, with optimal weights (requires knowledge of the test distribution; optimal importance weights derived in Appendix B)

- IRM (needs access to additional environment metadata)

Implementation details can be found in Appendix A.

## 4.1 EVALUATION

As discussed above, we evaluate our approach by training relatively simple models on the outputs of the frozen trained models to adapt to a shift in distribution that destroys spurious correlations in the training distribution. We consider the following adaptation models:

- **Best**: choose the single best performing model from the collection, and use its frozen predictions (0 parameters)

- **Linear**: train a logistic regression model (with $L^2$ regularization) on the concatenated frozen final layer representations (i.e. the outputs of each representation function $h_i$) from each model ($n \times D_{\text{repr}}$ parameters for $n$ models and representation dimension $D_{\text{repr}}$)

These adaptation models are trained on 500 observations (1% the size of training set) from the new distribution. We tune hyperparameters using a validation set drawn from the new test distribution and report results on a hidden test set corresponding to maximal validation scores achieved. We treat the number of models in our framework as a hyperparameter and present results for the best setting. We tune hyperparameters for IRM using the approach used to produce the results in Arjovsky et al. (2019). Results for each experimental condition are averaged over 10 runs, for which we report the standard deviation. For baselines requiring multiple models, we report results for 10 models as adding further models did not improve results. For single-model baselines, the 'best' protocol computes the test accuracy for that single model, while the 'linear' protocol computes the test accuracy for a linear classifier trained on the frozen output of the single representation function.

## 4.2 INTERPRETATION OF RESULTS

For the C-MNIST benchmark (Table 1), our approach outperforms all baselines under the 'best' evaluation protocol. Results under this evaluation protocol indicate that the approach has successfully isolated the 'digit' signal into a single model to the extent that, given this model, no further fine-tuning is required to make accurate test time predictions. Across all experimental conditions, our approach also produces a single model in the collection exploiting the spurious colour signal, evidenced by the corresponding model's maximal training set performance and chance performance on the test distribution. The latter is further evidence of our framework successfully isolating predictive signals into distinct models.

As expected, the single ERM classifier baseline fails across all experimental conditions. Single model baselines with regularization that diversifies the components of the single model's representation (i.e. DeCov, MA, and UER) often outperform the ERM baseline but are not competitive with our approach. Perhaps surprisingly, the ERM baselines with multiple models as well as the 'pairwise cosine' baseline are competitive with our approach under the 'linear' protocol, even though no single frozen model from each respective collection can make accurate test-time predictions (as evidenced by the results under the 'best' protocol). This suggests that the superficial notions of model diversity used in these cases are sufficient, given a large enough number of models, to allow for each model to retain different, small amounts of information about the 'digit' signal such that, taken together, all the small sources of information are sufficient to make accurate test-time predictions. While we acknowledge that this may just be an artefact of the benchmarks considered in this work, we claim that exploring this phenomenon further in more diverse settings is an interesting direction for future work.

The conditional estimator outperforms the unconditional version across all experimental conditions.

Table 1: Test accuracies for the C-MNIST task. Test data is drawn from a distribution in which the colour signal has been destroyed.

|  | Test accuracy (s.e.) MNIST | | Test accuracy (s.e.) Fashion-MNIST | |
| --- | --- | --- | --- | --- |
|  | Linear | Best | Linear | Best |
| Optimal | 75 | 75 | 75 | 75 |
| Optimal IWERM | 70.3 (0.6) | 70.7 (0.5) | 70.2 (0.7) | 70.5 (0.5) |
| IRM (requires env. info.) | 69.7 (1.1) | 68.6 (1.2) | 69.0 (0.5) | 68.4 (0.6) |
| Ours (conditional, 2 models) | 69.8 (0.4) | 69.8 (0.4) | 70.0 (0.7) | 70.2 (0.4) |
| Ours (unconditional, 2 models) | 66.7 (1.0) | 66.6 (0.9) | 68.4 (1.1) | 68.0 (0.6) |
| ERM (single model) | 56.8 (2.6) | 50.3 (0.7) | 61.4 (2.9) | 51.4 (0.5) |
| DeCov (single model) | 55.6 (3.4) | 54.5 (1.1) | 57.7 (4.2) | 52.9 (0.8) |
| MA (single model) | 63.7 (2.5) | 51.0 (0.5) | 60.3 (1.2) | 50.7 (0.5) |
| UER (single model) | 57.3 (4.7) | 51.2 (0.5) | 57.1 (2.3) | 50.7 (0.6) |
| ERM (10 models, ensemble) | 67.4 (1.0) | 51.0 (0.8) | 67.6 (1.5) | 50.5 (0.6) |
| Pairwise cosine (10 models) | 70.5 (0.8) | 56.4 (0.8) | 69.5 (0.5) | 54.4 (1.1) |

Table 2: Test accuracies for the RC-MNIST task. Test data is drawn from a distribution in which the colour signal has been destroyed.

| Method | Test accuracy (s.e.) MNIST | | Test accuracy (s.e.) Fashion-MNIST | |
| --- | --- | --- | --- | --- |
|  | Linear | Best | Linear | Best |
| Optimal | 75 | 75 | 75 | 75 |
| Optimal IWERM | 72.0 (0.6) | 72.2 (0.5) | 70.9 (0.6) | 70.9 (0.4) |
| IRM (requires env. info.) | 70.2 (0.9) | 70.3 (0.5) | 69.2 (0.4) | 69.5 (0.5) |
| Ours (conditional, 2 models) | 71.3 (0.7) | 71.4 (0.8) | 70.5 (0.5) | 70.4 (0.7) |
| Ours (unconditional, 2 models) | 64.0 (2.6) | 65.3 (1.3) | 68.1 (1.0) | 68.9 (0.5) |
| ERM (single model) | 61.6 (7.1) | 50.4 (0.4) | 69.7 (1.1) | 52.0 (1.6) |
| DeCov (single model) | 64.3 (3.2) | 52.4 (1.6) | 67.5 (2.0) | 51.1 (0.9) |
| MA (single model) | 70.2 (1.4) | 51.4 (0.5) | 69.5 (1.2) | 51.3 (0.7) |
| UER (single model) | 70.9 (1.2) | 51.4 (1.0) | 69.5 (1.0) | 51.2 (0.7) |
| ERM (10 models, ensemble) | 71.5 (0.5) | 52.5 (1.5) | 70.1 (0.4) | 51.3 (0.9) |
| Pairwise cosine (10 models) | 71.8 (0.4) | 53.2 (0.8) | 70.7 (0.4) | 51.1 (0.6) |

Table 3: Test accuracies for the TC-MNIST task. Test data is drawn from a distribution in which the two colour signals have been destroyed.

| Method | Test accuracy (s.e.) MNIST | | Test accuracy (s.e.) Fashion-MNIST | |
| --- | --- | --- | --- | --- |
|  | Linear | Best | Linear | Best |
| Optimal | 75 | 75 | 75 | 75 |
| Optimal IWERM | 69.2 (0.8) | 61.7 (2.0) | 68.9 (2.1) | 61.3 (1.9) |
| IRM (requires env. info.) | 67.5 (1.3) | 62.8 (0.8) | 68.8 (0.7) | 65.5 (0.7) |
| Ours (conditional, 3 models) | 67.5 (0.7) | 67.8 (0.8) | 67.9 (0.8) | 67.6 (2.4) |
| Ours (unconditional, 3 models) | 59.0 (1.4) | 57.6 (2.3) | 66.8 (0.7) | 64.1 (3.4) |
| ERM (single model) | 54.7 (3.7) | 56.3 (0.5) | 60.8 (2.8) | 59.4 (0.6) |
| DeCov (single model) | 62.0 (1.6) | 59.1 (0.9) | 63.4 (3.3) | 59.6 (0.6) |
| MA (single model) | 58.8 (3.9) | 58.1 (0.7) | 60.6 (2.6) | 59.3 (0.7) |
| UER (single model) | 62.6 (1.9) | 58.2 (0.9) | 65.7 (3.5) | 59.3 (0.5) |
| ERM (10 models, ensemble) | 68.6 (1.6) | 58.5 (1.1) | 69.0 (0.9) | 58.7 (1.0) |
| Pairwise cosine (10 models) | 68.2 (1.1) | 59.2 (0.7) | 67.0 (1.2) | 59.7 (0.6) |

We observe a similar pattern of results for the RC-MNIST benchmarks (Table 2). For our framework, this is an indication that the framework is robust to the obstacle to extracting conditionally independent predictive signals entailed in this benchmark.

A similar pattern of results emerges for the TC-MNIST benchmarks (Table 3), with some exceptions: relative to the other benchmarks, performance drops for the optimally importance-weighted baseline. We have no explanation for this other than to defer to work pointing out potential issues with importance weighting when training overparameterized deep neural networks (Byrd & Lipton, 2019).

We note that our approach is competitive with IRM in isolating the causally correct predictive signal across all experimental conditions, despite having no access to the environment metadata needed by the former to identify invariant predictive signals.

## 5  RELATED WORK

Our work shares motivation with Quality-Diversity (QD) algorithms (Pugh et al., 2016) from the evolutionary computation literature, which also attempt to find diverse, high-performing solutions to a task. This is typically done by partitioning the search space using a predefined behaviour descriptor and finding high-performing solutions within each behavioural niche (Mouret & Clune, 2015; Cully & Demiris, 2017). We do not explicitly partition the search space but instead enforce diversity through an information theoretic objective. A central challenge in directly applying existing QD algorithms to the problems considered in this work is the difficulty in defining an appropriate behaviour descriptor. An interesting direction for future work would be to investigate the possibility of learning suitable behaviour descriptors from data, as explored in the context of control problems in Cully (2019).

Ensemble-based methods (Polikar, 2006; Rokach, 2010) train collections of models using various techniques (e.g. subsampling of observations and features (Breiman, 2001), boosting (Friedman, 2001), penalizing the correlation of model predictions (Liu & Yao, 1999)) to enforce some notion of model diversity such that aggregate predictions outperform single model predictions. While we have included ensemble baselines, none of the aforementioned approaches are appropriate for the setting considered in this paper.

Various approaches, which we consider as baselines, use regularization intended to diversify some aspect of the model. Yu et al. (2011) minimize cosine similarity between weight vectors of output components. Cogswell et al. (2015) penalize the magnitude of non-diagonal entries of the covariance matrix of activations. Xie et al. (2015) enforce large mean angles between the weight vectors, minus the variance of the angles. Xie et al. (2017) enforce eigenvalues of the matrix of weight vectors to be uniform, hypothesized to yield uncorrelated and evenly useful components. We note that diversifying the components of a single model without requiring that each are separately sufficient to produce good predictions is susceptible to a scenario in which learned components are distinct but only some are predictive. Our approach precludes this scenario by construction.

A number of works (Kim & Mnih, 2018; Chen et al., 2018; Poole et al., 2019; Gao et al., 2019) use total correlation based objectives to enforce factorized representations in generative models. Similar to our approach, FactorVAE (Kim & Mnih, 2018) uses adversarial training to minimize an estimator of total correlation. In contrast to these approaches, we target total correlation for a collection of high-dimensional vector-valued representations while also leveraging label information. A possibility for future work is to impose probabilistic structure on the representations in our proposed method that may enable the use of the upper bound on total correlation presented in Poole et al. (2019), perhaps alleviating the need for adversarial training.

Independent component analysis (ICA) (Comon, 1994) addresses the problem of separating a signal into distinct components using an independence based criterion. This is done in an unsupervised setting and making different assumptions than we do. Brakel & Bengio (2017) use an adversarial training method similar to ours in the context of non-linear ICA.

Other works (Kwok & Adams, 2012; Mariet & Sra, 2015; Pang et al., 2019; Parker-Holder et al., 2020) use constructs derived from determinantal point processes (Kulesza & Taskar, 2012) to com-

pute diversity measures. These are not directly applicable to our setting, but exploring these approaches in our context is an interesting direction for future work.

There is a large body of work attempting to account for test-time distribution shift (Quionero-Candela et al., 2009). The problems considered in this paper are an example of covariate shift, in which the marginal distribution of the label remains unchanged at test time. Approaches from the literature to address covariate shift are based on discarding input features, reweighting training observations, or domain adversarial training (Ganin et al., 2016). Approaches based on discarding input features are not readily applicable to our context considering high-dimensional unstructured inputs. We have included results for an optimally importance-weighted baseline and have refrained from exploring baselines which estimate the weights from data due to the challenges entailed in doing so in a deep learning context. We were unable to train a domain adversarial approach to produce good results under the evaluation protocols considered in this paper; we hypothesize that this may be due the scarcity of adaptation data under our evaluation protocols. We have deliberately refrained from considered standard image domain adaptation benchmarks as these lie outside of the intended scope of our method.

Some works from the GAN literature (Hoang et al., 2018; Ghosh et al., 2018) attempt to prevent mode collapse by training a collection of generators to produce easily discriminable samples. Experiments adapting such approaches to our context fail completely and we have excluded corresponding results. This is not surprising: in our setting it is easy for learned representations to be easily discriminable while still representing the same information content.

Other works drawing from ideas in the causal inference literature attempt to yield models that generalize better under certain violations of the ERM assumptions. Invariant Risk Minimization (Arjovsky et al., 2019) assumes access to data from multiple environments and attempts to find a representation such that the same classifier is simultaneously optimal in all environments (as do subsequent works addressing the same / similar problem (Ahuja et al., 2020; Krueger et al., 2020)). The problem of causal discovery (Pearl, 2009; Peters et al., 2017) consists in trying to identify underlying causal structure from data such that the tools of causal inference can then be used to compute predictions that are valid under certain shifts in distribution.

## 6 CONCLUSION

We have introduced a new method for training a collection of high-performing supervised learning models each incentivized to learn a distinct way to solve the task. We have empirically demonstrated how minimizing our proposed measure of model diversity in addition to the empirical risk suffices to yield a collection of models exploiting distinct predictive signals present in benchmark tasks.

Perhaps surprisingly, we find that simple notions of model diversity considered in certain baselines are sufficient to produce competitive results under a linear fine-tuning evaluation protocol. While this may just be an artefact of the benchmarks considered in this work, we claim that such a strategy (i.e. simple, scalable measures of model diversity coupled with linear fine-tuning given small amounts of new data) is an interesting avenue to explore in future work.

We have focused on evaluating our approach on datasets which we have sufficient control over to enable the testing of basic hypotheses about our proposed framework. To the best of our knowledge, there are currently no other more complex, natural datasets for us to evaluate our approach on. Furthermore, as in Arjovsky et al. (2019), we require access to evaluation data from a different distribution than the training data in order to tune hyperparameters. Rather than seeing this as a disadvantage of our approach, we see it as further evidence that confronting the problem of generalization in a data-driven way may call for the development of new kinds of datasets and benchmarks that are better suited to testing robustness to violations of the i.i.d. assumption on which the empirical risk minimization principle is founded.

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

## A  IMPLEMENTATION DETAILS

In the notation of Section 3, we use multilayer perceptrons (MLP) as representation functions $h_i$ with two hidden layers of size 128 and 64 respectively, 32 output units, and ReLu activations. We use the cross-entropy as a classification loss. Our single ERM classifier baseline is equivalent to a single model from the collection of models in our framework. The baselines calling for multiple models use multiple instantiations of this same model with different initial weights. The variational critic $f$ is an MLP with 2 hidden layers of size 256 computing a scalar output from an input consisting of a concatenation of all representations computed by the $h_i$'s. We use Leaky ReLus with a slope of 0.2 as activations in the critic as this appeared to speed up training in experiments. We use $\beta = 10$ throughout. For the 'pairwise cosine', we use a coefficient of 0.1 for the loss function term computing the average pairwise cosine distance between final layer representations for a given input.

We use RMSProp with a learning rate of $10^{-5}$ as an optimizer for the models in the collection as well as the critic. We note that not using a momentum-based optimizer appears to be crucial for stable training, following the trend of decreasing momentum values in the GAN literature (Gidel et al., 2019). We note that training appears to be further stabilized by normalizing representations computed by the $h_i$'s to have unit $L^2$ norm prior to being input into the critic.

We use a batch size of 256 for training both the models and the critic, as well as $M = 64$ in Equations (6) and (7). To train the simple classifiers, we use 500 training and validation observations from the novel test distribution, the latter used to tune the hyperparameters (if any) of the newly trained classifiers. We report test accuracies computed for the remaining 9000 test observations. We train for 250 epochs and report test accuracies corresponding to maximal validation accuracies. We make the following exceptions for the TC-MNIST task: we train for 1000 epochs and alternate between 5 steps of training the critic and 1 step of training the collection of models. This would appear to be necessary for obtaining useful total correlation gradients given the relatively more challenging variational problem that needs to be solved in this case.

These hyperparameter settings were determined from manual tuning of our approach on the C-MNIST benchmark solely for the two-model case using the conditional TC estimator, and reused across all other experimental conditions. Minimal manual tuning of the regularization coefficient for the 'linear' evaluation protocol was subsequently performed for each experimental condition with a distinct number of models in the collection. Coefficients for any regularization terms in baselines were similarly tuned.

Hyperparameters for IRM were chosen using the tuning method used to produce the results in Arjovsky et al. (2019), owing to the similarity of the datasets considered. Hyperparameters were chosen using random search with 50 trials, sampling values as in `https://github.com/facebookresearch/InvariantRiskMinimization/blob/master/code/colored_mnist/reproduce_paper_results.sh`. The best model was chosen on the basis of the mean final test accuracy (over 10 runs) under each of our evaluation protocols.

### A.1  DATA SYNTHESIS

#### C-MNIST

The Colored MNIST dataset is constructed as in Arjovsky et al. (2019).[2] MNIST images are sub-sampled to 14x14 size, images are assigned to one of two classes (depending on whether the MNIST label is 0-4 or 5-9), and this binary class label is corrupted by flipping with probability 0.25. Subsequently, a binary colour signal is computed from the corrupted label by flipping it with probability

---

[2]Based on code from: `https://github.com/facebookresearch/InvariantRiskMinimization`

$p_e$, where $e$ is an environment index. The colour signal is represented in the final dataset by appending a new channel to the grayscale image. The new channel is set to a copy of the grayscale image if the color signal is equal to 1, and zeroed out otherwise. The final dataset consists of these two-channel 'images' and the corrupted label. Two training environment with $p_{e_1} = 0.2$ and $p_{e_2} = 0.1$ are used to train the IRM baseline. Our models and the ERM baseline are trained by collapsing these two training environments into one, discarding the environment information.

## RC-MNIST

There are a few key changes in synthesizing the Rotated Colored MNIST dataset relative to C-MNIST. Firstly, for each observation a new 'common cause' variable is sampled from Bernoulli(0.5). Subsequently, the digits in the C-MNIST pipeline are rotated by 45 degrees if the common cause variable is equal to 1, and are not rotated otherwise. The color signal is represented differently than in C-MNIST. Instead of copying the grayscale image in a new channel, the new channel is set to a constant value of 1.0 across all spatial entries if the color signal is equal to 1, and is zeroed out otherwise. Finally, this color channel is shifted up by 0.1 if the common cause variable is equal to 1, and is shifted down by 0.1 otherwise to produce the final dataset.

## TC-MNIST

Two-Colored MNIST is similar to RC-MNIST (but without the perturbations due to the common cause variable), with the addition of a new channel representing an additional color signal computed from the corrupted label. For each observation, this new color signal channel is perturbed by sampling a single value from Normal(0, 0.1) and adding the value to all spatial entries in the channel.

For the purposes of training the IRM baseline, the new color signal is computed as for the original signal but using a new environment specific corruption parameter. Similarly, two environments are respectively created using environment parameters of 0.2 and 0.1 for the original color signal, and 0.3 and 0.2 for the new color signal. Both environments are collapsed into one for training our models and the ERM baseline.

## B    DERIVATION OF OPTIMAL IMPORTANCE WEIGHTS

Assuming that the supports of the training and test distributions overlap, it can be shown (Quionero-Candela et al., 2009) that the following importance weights for the risk under the training distribution produce an unbiased estimator of the risk under the test distribution:

$$w^*(X, Y) := \frac{p_{test}(X, Y)}{p_{train}(X, Y)} \tag{9}$$

where $p_{train}$ and $p_{test}$ denote training and test distributions respectively.

For the benchmarks considered in this paper, we have access to the complete generative model that produces the data and are thus able to compute these optimal weights in closed form (assuming the input $X$ is computed from the raw digit image $D$ and color signal $C$). For C-MNIST:

$$
\begin{aligned}
w^*(X, Y) \quad :=& \frac{p_{test}(X, Y)}{p_{train}(X, Y)} \\
=& \frac{p_{test}(Y)p_{test}(X|Y)}{p_{train}(Y)p_{train}(X|Y)} \\
=& \frac{p_{test}(X|Y)}{p_{train}(X|Y)} \quad \text{(since marginal distributions of } Y \text{ are the same)} \\
=& \frac{p_{test}(C, D|Y)}{p_{train}(C, D|Y)} \\
=& \frac{p_{test}(C|Y)p_{test}(D|Y)}{p_{train}(C|Y)p_{train}(D|Y)} \quad \text{(since colour and digit are conditionally independent given } Y) \\
=& \frac{p_{test}(C|Y)}{p_{train}(C|Y)} \quad \text{(since conditional distributions of digit given label are the same)} \\
=& \frac{p_{test}(C)}{p_{train}(C|Y)} \quad \text{(since colour is independent of label at test time)} \\
=& \frac{0.5}{p_{train}(C|Y)} \quad \text{(under test distribution colour is Bernoulli(0.5))} \\
=& \frac{0.5}{0.85} \text{ if } C = Y \text{ else } \frac{0.5}{0.15}
\end{aligned}
\tag{10}
$$

That is, observations where colour and label do not coincide (i.e. contradicting the spurious correlation in the training distribution) are relatively upweighted.

