# OpenReview forum: "Learning a Non-Redundant Collection of Classifiers"
_ICLR.cc/2021/Conference — Reject_

### Official Review · AnonReviewer2 · 2020-10-27
**Interesting goal, but not entirely clear why, theoretically, the method should work.**

**Rating:** 4
**Confidence:** 3

**Review:**

The paper discusses the problem of inducing complementary representations to improve classification and isolate relevant signals among possible spurious/artificial ones. This is certainly an interesting agenda, and the experiments seem to provide some empirical evidence that the proposed method seems to work reasonably well, yet I must say that from the theoretical side, I did not manage the identify the reasons for this. Also, the paper is sometimes difficult to follow, as it is not totally rigorous and also sometimes assume strong background knowledge about very specific points.

Main remarks:

* While the announced agenda of the authors seems to get rid of spurious correlations in the original features and to identify informative, non-redundant signals issued from those, the whole paper concerns possibly complex transformations h_n of those (by the way, has the number of functionals to be the same as the one of features?). How does this compare to, say, find an efficient Binary Coding Matrix with low overlap or with feature selection approaches?

* P4, top: the estimator is built using an analogy with the pairwise case, however nothing is said about the properties of this analogy. Does it remain a lower bound? How good an approximation of the original goal can we expect it to be?

* P4, top half: a lot of sampling needs to be done to get the estimates (again, are we speaking of X_i or h_i(X)? Since we are in the same section as equation (5), why Y has now turned into what seems to be the target classification variable?), how are they actually achieved, since most of them require access to theoretical distributions?

* The results are a bit disappointing when allowing for linear combination of the classifiers, and from the description it is unclear to me whether Optimal IWERM is an obtainable model or not (I understand that it is, but it is not explicitly mentioned in the list, or is the IWERM the last model mentioned at the bottom of page 4? And in this case what is "optimal")? Because it performs quite comparably to the proposed model in the Best case. Also, why is there a difference between linear combination of models and best models in the case of ERM (single model)?


Minor remarks:

* P2, it should be said that KL is for Kullback-Leibler divergence

* P3, top, what is k? Are c_i vectors or functions (from where to where in the last case)? The composition dot symbol suggest the latter.

* P3, top: are \theta parameters of h_i or c_i?

* P3, before equation (5): suddenly Y becomes a feature and not the target class variable. This is confusing at first.

* P3, bottom: what is a differentiable variational critic computing unnormalised score? Can a reference be given?

* P4, bottom: "and only included for comparison" --> "and are only included for comparison"

* P5, bottom: "we tune ... drawn from the new test distribution". There is unclarity about whether the test distribution is a subset of the training set or data issued from a test set (in which case certainly one cannot tune hyper-parameters using them). Maybe a synthetic drawing would help to understand how are created/used the noisy data sets?

---

> ### Author Response · Authors · 2020-11-17
> **Author response to reviewer 2**
>
> Thank you for your time and detailed feedback. We will send an additional update within the discussion period when any promised revisions have been made.
>
> > “the experiments seem to provide some empirical evidence that the proposed method seems to work reasonably well, yet I must say that from the theoretical side, I did not manage the identify the reasons for this.”
>
> We think that there is value in communicating the success of the method given the potential deviation from any theoretical expectations (which we can only speculate about at this stage as this paper is intended only to document empirical investigations).
>
> > (Clarification: goal of paper) “the announced agenda of the authors seems to get rid of spurious correlations in the original features and to identify informative, non-redundant signals issued from those”
>
> The goal of the paper is to learn vector-valued representations of distinct predictive signals present in a training set of unstructured data, with respect to a measure of distinctness based on information theoretic redundancy. Our experiments show that, as a consequence of enforcing our chosen measure of diversity, under our evaluation protocols we are able to isolate spurious and robust signals into distinct models.
>
> > “has the number of functionals to be the same as the one of features?”
>
> There is no requirement that the number of models be the same as the dimensionality of the representation computed by h_i. In principle, each h_i could compute a vector of a different size, although this may perhaps impact the variational problem in some way; in our implementation each h_i computes a vector representation of the same size across models. The number of models is a hyperparameter.
>
> > “How does this compare to, say, find an efficient Binary Coding Matrix with low overlap”
>
> Please could you provide a reference for “finding an efficient Binary Coding Matrix” so we can make accurate comparisons?
>
> > “or with feature selection approaches?”
>
> A central difference between feature selection methods and our approach is that we focus on unstructured data (images in our benchmarks) and jointly consider the problem of learning appropriate features from such data. Also the emphasis is on jointly optimizing distinctness and predictive power of features in our approach.
>
> > (On theoretical analysis) “nothing is said about the properties of this analogy. Does it remain a lower bound? How good an approximation of the original goal can we expect it to be?”
> The main goal of the paper is to communicate the empirical findings emerging from our proposed construction. Theoretical analysis of our estimator and construction of alternatives is something we have deferred to future work. If it turns out that we are in fact minimizing a lower bound we speculate that our method may still provide a useful learning signal. Intuitively, minimizing the value computed by the variational estimator amounts to optimizing representations to reduce the variational critic's ability to discriminate between samples from the joint and product of marginals, a proxy for small total correlation between the representations of each model (https://arxiv.org/pdf/0809.0853.pdf). The method’s success may also be attributable to our strategy of only applying a small number of weights update for the representation functions before updating the variational estimator.
>
> > (Clarification: computation of estimator) “a lot of sampling needs to be done to get the estimates (again, are we speaking of X_i or h_i(X)? Since we are in the same section as equation (5), why Y has now turned into what seems to be the target classification variable?), how are they actually achieved, since most of them require access to theoretical distributions?”
>
> The estimator is always computed on the representations, i.e. the outputs of the h_i functions, not the raw input. For the proposed conditional estimator, we partition a batch based on the label Y and separately compute the estimator for each subgroup, finally taking a weighted average to compute our proposed conditional estimator. The weights are estimated using the frequency of each label in the batch. Thank you for bringing this to our attention, we will clarify this in a revision.
>
> > (Clarification: linear protocol) “results are a bit disappointing when allowing for linear combination of the classifiers”
>
> It’s important to note that the 'linear' protocol is not based on a linear combination of the classifiers, but a linear model trained on the concatenated frozen outputs of the representation functions of different models. We have excluded results for an evaluation protocol where a linear model is trained on the frozen classifier outputs to simplify presentation since these results were not particularly interesting: as expected the baselines here underperform by merit of discarding more information relative to the ‘linear’ protocol. We will add a clarifying comment in a revision.

---

> ### Author Response · Authors · 2020-11-17
> **Author response to reviewer 2 (continued)**
>
> > “it is unclear to me whether Optimal IWERM is an obtainable model or not”
>
> 'Optimal' refers to the optimal importance weights that we are able to derive analytically given closed form access to the test distribution. The derivation is given in the Appendix B. As such and as stated in the paper, this method is merely included as a yardstick, not as a fair comparison to our method and the rest of the baselines that assume access to the same information (i.e. a small number of samples from the test distribution).
>
> > “why is there a difference between linear combination of models and best models in the case of ERM (single model)?”
>
> Under the 'linear' protocol a linear classifier is retrained on the output of the representation function, not the scalar output of the classifier.
>
> > “what is a differentiable variational critic computing unnormalised score? Can a reference be given?”
>
> This is borrowing terminology used for variational estimators of mutual information in a deep learning context (e.g. https://arxiv.org/pdf/1905.06922.pdf). We will add this to a revision submitted during the discussion period. In our setting, it is just a differentiable function of concatenated vector representations computing a real number.
>
> > “There is unclarity about whether the test distribution is a subset of the training set or data issued from a test set”
>
> The test set is indeed drawn from a different distribution than the training set. The emphasis of our evaluation protocols is on our method's success relative to the baselines when only having access to a small amount of data from the test distribution. We note that other works such as IRM use a similar oracle method for hyperparameter tuning. A model selection strategy that does not need such data is not something we rule out as a possibility and is a central priority for future work. We currently see no alternative means to evaluate our approach within the rigid constraints of the problem setting. If we compare to the model selection strategies used for the related but distinct problem of domain generalization, methods in the literature propose methods also based on oracle data or alternatively using held out training domains as validation sets (https://arxiv.org/pdf/2007.01434.pdf). We deliberately assume no such access to domain information in our problem setting (our goal is distinct to that of domain generalization and we would argue may not require data from different domains) and hence cannot apply similar methods. Ultimately though, the results presented on the basis of our current evaluation protocol do provide evidence that our framework is able to outperform baselines wrt our stated goal of isolating signals that are distinct wrt out proposed measure of diversity.
>
> > (Other minor remarks)
>
> Thank you, we will address all of the other minor remarks in a revision.

---

### Official Review · AnonReviewer1 · 2020-10-28
**Intuitive and simple approach but experiments + related work are somewhat weak**

**Rating:** 4
**Confidence:** 4

**Review:**

This paper proposes an approach of training an ensemble of DNN classifiers while also minimizing the total correlation (TC) between the last layers (learned feature representations) of the classifiers to increase robustness to spurious correlations.  To compute gradients for this new TC regularization term, they use the InfoNCE objective as a proxy, and minimize this, and then use alternating minimization to learn the parameters. The authors then test this on a variation of the Colored MNIST task, showing gains over baseline approaches.

Overall, this is a simple and intuitive approach.  Simple and intuitive is good- but requires (A) rigorous comparison to prior work to show why something like this has not been done before, and (B) rigorous experimental comparison to other approaches.  Unfortunately this paper does not do enough here.

Regarding related work- the idea of adding a term that optimizes for diversity of the learned features is not a new one (random sample of papers: https://openreview.net/pdf?id=Hy1QdPyvz, https://www.researchgate.net/publication/330257555_Regularizing_Deep_Neural_Networks_by_Enhancing_Diversity_in_Feature_Extraction, http://www.cs.cmu.edu/~epxing/papers/2017/Xie_Singh_Xing_ICML17.pdf; and many others).  Note in particular that while the authors discuss an "ensemble" of classifiers, there is not a major distinction between this and a single model where the diversity of that model's features is regularized.  The authors here compare to a wide range of approaches- QD algorithms, ensemble-based methods, factorized generative models, ICA, and more- but seem to miss more basic comparables.

More importantly: the experiments compare to only one other method that is explicitly not a valid apples-to-apples comparison because of different input information required; and only examine one fairly bespoke semi-synthetic dataset.  The authors would need to compare to (A) a broader range of similar feature diversity objective approaches, and (B) should have explored whether this actually works on standard real-world datasets (where spurious correlations naturally occur), if they wanted compelling and general results.

---

> ### Author Response · Authors · 2020-11-17
> **Author response to reviewer 1**
>
> Thank you for your time and detailed feedback. We will send an additional update within the discussion period when any promised revisions have been made.
>
> > “the idea of adding a term that optimizes for diversity of the learned features is not a new one”
>
> Thank you for the additional references, we will include them in a revision. As noted in the related work, approaches based on negative correlation of predictions in an ensemble (e.g. https://openreview.net/pdf?id=Hy1QdPyvz) are inappropriate in our setting (the distinct signals in our benchmarks are correlated by construction) hence our exclusion of such baselines. (https://www.researchgate.net/publication/330257555_Regularizing_Deep_Neural_Networks_by_Enhancing_Diversity_in_Feature_Extraction) utilizes an approach based on cosine similarity minimization very similar to one of our baselines. We will attempt to include results using http://www.cs.cmu.edu/~epxing/papers/2017/Xie_Singh_Xing_ICML17.pdf as a baseline in a revision, although this is not directly amenable to the multiple model setup we consider. We will add further discussion to the related work and attempt to add further baselines emerging from this discussion in a revision.
>
> > “there is not a major distinction between this and a single model where the diversity of that model's features is regularized”
>
> We claim that there is a distinction between these two approaches for the reason that e.g. satisfying a disentangling constraint for a single model can be done without learning to represent a variety of *predictive* signals. For example, in Colored MNIST a solution in which some representation capacity pertains to the spuriously predictive 'color' factor of variation while the rest is dedicated to factors of variations that are disentangled but not predictive is optimal on the training set. However, this is not aligned with the goals set out in this paper. By requiring that representations are separately sufficient to solve the task, our approach eliminates the possibility of this situation and hence we would argue it is different in an important way.
>
> > “the experiments compare to only one other method that is explicitly not a valid apples-to-apples comparison because of different input information required...authors would need to compare to (A) a broader range of similar feature diversity objective approaches”
>
> IRM is merely included to compare to what a state-of-the-art (https://arxiv.org/pdf/2007.01434.pdf) method for a related problem is able to achieve with access to additional information that our method does not have access to. We are explicit in stating that it is not a like-for-like comparison. We have argued in the related work for the exclusion of certain other diversity objectives as baselines due to their inappropriateness for our problem setting. We will include results for additional baselines emerging from this discussion in a revision.
>
> > “should have explored whether this actually works on standard real-world datasets (where spurious correlations naturally occur)”
>
> Finding appropriate benchmarks has stood out as the main challenge of this work. Short of having a model selection strategy based on data drawn solely from the training distribution, we have focused on an evaluation protocol that requires a small amount of test data from a shifted distribution and have accordingly evaluated our approach's merits relative to baselines. What has prevented the use of more real-world datasets known to have spurious correlations is a lack of appropriate test sets for use with our evaluation protocols. The closest we have come across are the robust vs non-robust test sets of (https://papers.nips.cc/paper/2019/file/e2c420d928d4bf8ce0ff2ec19b371514-Paper.pdf) for CIFAR10. We have some promising preliminary results here with respect to our diversity measure's ability to discriminate between the robust and non-robust signals in this setting. We have excluded discussion of this pending more thorough investigation and other subtleties that may preclude these from being suitable test sets for our setting.
>
> We think that the work has value to community in the form of: empirical success relative to the baselines and oracle solutions, some element of surprise in the success of a method which may involve minimizing a lower bound, favourable comparisons to state-of-the-art approaches which are able to exploit information that our approach does not have access to, as well as perhaps some surprise at success of the baseline measures of diversity under our 'linear' evaluation protocol which to our knowledge has not been similarly demonstrated in this context.

---

### Official Review · AnonReviewer4 · 2020-10-30
**Interesting idea, experimental results and analysis need to be strengthened**

**Rating:** 5
**Confidence:** 4

**Review:**

This paper proposes to learn a collection of classifiers, each of which is incentivized to use distinct features. The motivation is that among these distinct models, some would leverage robust signals present in the training data that can be generalized to test-time distribution shift; in contrast to ERM training where a model will use a mixture of robust and spurious signals and thus fail to generalize.

In particular, the authors train multiple models that minimize both their ERM losses and the total correlation (TC) of their final layer representations conditioned on the label. TC is an extension of mutual information (MI) that considers n random variables, and the authors similarly extend the InfoNCE estimator of MI to n variables to estimate TC.

Experiments are conducted on the MNIST and Fashion-MNIST datasets, with three types of spurious signals injected into training data. Test data is constructed such that the spurious correlations no longer exist. Methods are tested in two adaptation setups: choosing the single best performing model from the collection, or training a linear model on top of the final layer representations from each model.

Overall, I think the idea of training a set of distinct models to separate robust and spurious features is interesting, and results under the "best" setup show that the proposed method has isolated a single model that can generalize to the test data, which is promising. However, under the widely adopted "linear" adaptation setup, the proposed method has no advantage over a simple ensemble or a collection of classifiers trained to maximize pairwise cosine distances between final layer representations, which makes the value and practicability of the proposed method unclear. In addition, can you provide analysis of which features are learned by each model in the collection? I think this is important for understanding the model behavior.

About TC estimation. InfoNCE is a lower bound of MI. In representation learning where one wants to maximize the MI between the feature representation and the label, it makes sense to maximize a lower bound. But here you want to minimize TC, why you choose a lower bound instead of an upper bound? The choice of lower bound also forces you to use a minmax objective, which might be a roundabout way.

Lastly, I think it would be interesting if there is a way to identify the robust models from the spurious ones (perhaps based on some inductive bias) without having to access the test distribution. This would also show the superiority of your method over the ensemble baselines that require fine-tuning on test examples.

---

> ### Author Response · Authors · 2020-11-17
> **Author response to reviewer 4**
>
> Thank you for your time and detailed feedback. We will send an additional update within the discussion period when any promised revisions have been made.
>
> > “However, under the widely adopted "linear" adaptation setup, the proposed method has no advantage over a simple ensemble or a collection of classifiers trained to maximize pairwise cosine distances between final layer representations”
>
> We would argue that the results under the 'linear' protocol are either (1) an artefact of the benchmarks or (2) are suggestive of the utility of adapting simple ensembles using small amounts of new data (noted in Section 4.2). We are not aware of (2) being similarly demonstrated in the literature and hence see it as one of the contributions of the paper. We included the results under the 'linear' protocol mainly because we thought that this was an interesting demonstration of the information that an aggregate of models trained using simple measures of diversity retains about the input (even if this is not as 'disentangled' as for our framework). However, we would emphasize that the results under the 'best' protocol should be the central focus as they are best suited to testing our main hypothesis of isolating signals that are distinct wrt our measure of diversity.
>
> > “can you provide analysis of which features are learned by each model in the collection?”
>
> There is a lot we can deduce from the construction of the experiments about what features the model must be exploiting. Consider our method training a collection of two models on Colored MNIST. The 'best' protocol indicates that a single model is sufficient to perform well on the test set without further training. This model cannot be exploiting the 'colour' signal (it has no predictive power on the test set) and therefore must be exploiting the only remaining signal (i.e. the 'digit' signal). Such a model attains similar levels of performance on both the training and test sets by merit of exploiting the same signal in both, further evidence that it is not exploiting the colour on the training set (otherwise it is possible to attain better training set performance). Evidence of the other model exploiting the colour comes in the form of it's near optimal performance on the training set and chance performance on the test set. We have excluded results demonstrating the latter to simplify presentation but will add clarifying comments to a revision.
>
> > “But here you want to minimize TC, why you choose a lower bound instead of an upper bound? The choice of lower bound also forces you to use a minmax objective, which might be a roundabout way.”
>
> The main goal of the paper is to communicate the empirical findings emerging from our proposed construction. Theoretical analysis of our estimator and construction of alternatives is something we have deferred to future work. If it turns out that we are in fact minimizing a lower bound, we speculate that our method may still provide a useful learning signal. Intuitively, minimizing the value computed by the variational estimator amounts to optimizing representations to reduce the variational critic's ability to discriminate between samples from the joint and product of marginals, a proxy for small total correlation between the representations of each model (https://arxiv.org/pdf/0809.0853.pdf). The method’s success may also be attributable to our strategy of only applying a small number of weights update for the representation functions before updating the variational estimator.
>
> > “I think it would be interesting if there is a way to identify the robust models from the spurious ones (perhaps based on some inductive bias)”
>
> Finding a model selection strategy that does not utilize an oracle test set (albeit a small one) is a central focus for future work. Indeed, the working hypothesis that underlies this work is that there is some absolute sense in which distinct predictive signals are distinct and can be determined to be so from the training data alone (a related argument is made in https://arxiv.org/pdf/2009.00329.pdf). Our current evaluation protocol does however provide evidence of our approach's potential for extracting and isolating distinct signals in the benchmarks, relative to the baselines.

---

### Official Review · AnonReviewer3 · 2020-11-03
**A novel area of investigation, but the presentation could be strengthened**

**Rating:** 6
**Confidence:** 3

**Review:**

This paper proposes a method for training an ensemble of classifiers for problems where spurious correlations may be present.  The models in the ensemble are encouraged to learn conditionally disentangled representations.  It is suggested that this disentanglement will result in the spurious correlations being consigned to a subset of the models in the ensemble.  The ensemble will then be ready to quickly adapt to a new data distribution without the spurious correlations, via selection of a single model that does not depend on the spurious correlations, or via reweighting of model outputs.

The author proposes a loss function that combines the standard ERM loss (summed over ensemble constituents), with a variant of the InfoNCE objective that is proposed as an estimate of total correlation.  The total correlation estimator involves a learned variational critic function. The authors argue for the use of conditional total correlation in the supervised setting, since, without conditioning, the minimization of total correlation interferes with the supervised learning objective.

The empirical evaluations focus on three variants of the MNIST dataset where spurious correlations have been introduced.

The learning of disentangled representations is highly studied in the context of unsupervised learning, which is alluded to in the paper.  This work is novel in that it focuses on the utility of disentangled learning in supervised learning.  While the proposed method does not directly learn a single model that can discard spurious correlations (like IRM is designed to do when multiple environments are available), it outputs an ensemble where some a priori unknown subset of the models do ignore the spurious correlations.

At a high level, there is a similarity to ensembling methods such as random forests where correlation between constituent models is deliberately suppressed. However, the motivation is different in that case: individual tree models are high-variance, and averaging uncorrelated trees reduces the variance of the overall model. The authors may want to cite that literature, however.

The paper points to an interesting direction of research.  Overall there are some ways in which the authors could strengthen the case for their method, and make the presentation clearer.

*    My understanding is that InfoNCE provides a lower bound on mutual information.  Hence it makes sense to maximize it when the goal is to maximize MI.  In this case, minimizing TC would seem to require minimization of an upper bound.  Can the authors justify their use of the proposed estimator, perhaps with a more rigorous derivation that it actually is an effective bound on TC in the context of minimization?  If it is indeed a lower bound, why does minimizing a lower bound work?
*    The actual definition of the estimator is unclear to me.  I understand $K$ to be the batch size.  It is clear how to plug in to Equation 6 to calculate the unconditioned estimator for a single batch. However, Equation 7 takes an expectation conditional on the label.  How is this estimated?  Are we to assume that the contrastive examples are sampled from the subset of the batch that shares the same value of $Y$ as $Y_i$?  I do not believe this is addressed in the paper.
*    The argument for conditioning in Section 3 is reasonable; however, it is not backed up by concrete results.  I would agree that, without conditioning, the TC objective would conflict with the ERM objective.  However, I would think some balance would be reached where an unconditioned objective would still yield decently disentangled representations and models with decent accuracies.  Can the authors show a simple simulation that proves the superiority of the conditioned objective?  This becomes more important with a large number of classes, where estimating the conditioned TC becomes more challenging.
*    The setup for the empirical results is an interesting and appropriate one for the method proposed.  However, the authors do not argue for its practical relevance.  Are there real-world problems where this kind of rapid adaptation of ensembles is required?
*    On a similar note, the paper left me curious whether this disentangled learning is useful for the original supervised task. For example, analogous to random forests, does the disentangled model improve upon the performance of a single model in any datasets?
*    The results on the MNIST datasets are compelling.  By my reading, the most impressive thing is the “Best” performance for “ours”.  I think the results could be further strengthened by including a simulation example where the function form is fully known, and the internal mechanics of the proposed method could be more fully explicated.  For example, the simulation setup of the IRM paper could be borrowed.  Then, the authors could show whether one of the members of the ensemble actually learns the correct causal model.  Right now, there are some compelling results on MNIST, but it could be made clearer what the method is actually doing.
*    A minor point: On first reading, I found the results to be confusing, since ERM (single model) differed between “Linear” and “Best”.  It seems the difference is that for “Linear”, the top layer of the model is retrained, and for “Best” the model is unchanged.  The authors may want to clarify what “Linear” and “Best” refer to for a single model.

---

> ### Author Response · Authors · 2020-11-17
> **Author response to reviewer 3**
>
> Thank you for your time and detailed feedback. We will send an additional update within the discussion period when any promised revisions have been made.
>
> > (Theoretical analysis) “Can the authors justify their use of the proposed estimator, perhaps with a more rigorous derivation that it actually is an effective bound on TC in the context of minimization? If it is indeed a lower bound, why does minimizing a lower bound work?”
>
> The main goal of the paper is to communicate the empirical findings emerging from our proposed construction. Theoretical analysis of our estimator and construction of alternatives is something we have deferred to future work. If it turns out that we are in fact minimizing a lower bound we speculate that our method may still provide a useful learning signal. Intuitively, minimizing the value computed by the variational estimator amounts to optimizing representations to reduce the variational critic's ability to discriminate between samples from the joint and product of marginals, a proxy for small total correlation between the representations of each model (https://arxiv.org/pdf/0809.0853.pdf). The method’s success may also be attributable to our strategy of only applying a small number of weight updates for the representation functions before updating the variational estimator.
>
> > (Clarification: definition of estimator)
>
> Thank you for bringing this to our attention. We partition a batch based on the label and separately compute the estimator for each subgroup. Finally we take a weighted average to compute our proposed conditional estimator. The contrastive samples are indeed obtained from the same batch, as is done with InfoNCE. We will clarify this in a revision.
>
> > (On unconditional estimator) “I would agree that, without conditioning, the TC objective would conflict with the ERM objective. However, I would think some balance would be reached where an unconditioned objective would still yield decently disentangled representations and models with decent accuracies. Can the authors show a simple simulation that proves the superiority of the conditioned objective?”
>
> We do have results using the unconditional estimator which we decided to exclude to simplify presentation. You are right to suspect that this also works to some extent: using the unconditional estimator does outperform the baselines but requires a greater computational budget than the conditional version. We agree that the unconditional estimator could certainly be useful in settings where conditioning becomes challenging. We will add comments to this effect in a revision.
>
> > (Real-word applicability of evaluation protocol) “The setup for the empirical results is an interesting and appropriate one for the method proposed. However, the authors do not argue for its practical relevance. Are there real-world problems where this kind of rapid adaptation of ensembles is required?”
>
> For concrete examples of the real-world relevance of the evaluation protocol, one can look to typical applications of quality-diversity algorithms e.g. learning a diverse collection of robot gaits to rapidly adapt to damage on the basis of a small amount of policy data from a damaged robot (https://arxiv.org/pdf/1407.3501.pdf), although these are not trivially adaptable to the supervised learning setup we consider. The evaluation protocol we use may also be relevant in an online learning setting where temporal metadata may allow for defining appropriate adaptation sets. We also cannot rule out the possibility of a model selection strategy that does not require test data from an oracle (albeit a small amount). Our main goal is to characterize and find predictive signals that are, in some sense, different; learning to separately extract spurious and robust signals is a downstream, ultimate goal which we hypothesize will emerge as a consequence of optimizing an appropriate measure of predictive signal diversity. Perhaps the extent to which signals are different wrt a given measure of diversity can be reliably measured without oracle test data. This is a central line of investigation for future work.
>
> > “On a similar note, the paper left me curious whether this disentangled learning is useful for the original supervised task. For example, analogous to random forests, does the disentangled model improve upon the performance of a single model in any datasets?”
>
> Across all experimental conditions, all methods produce a single model attaining close to the maximal theoretical performance on the training set. We have excluded these results to simplify the presentation but will add clarifying comments in the revision. However, attaining good performance on the training set is not particularly interesting since exploiting the spurious signal by construction yields the best performance on the training set, and a single ERM model achieves this in a few gradient steps.

---

> ### Author Response · Authors · 2020-11-17
> **Author response to reviewer 3 (continued)**
>
> > “the simulation setup of the IRM paper could be borrowed. Then, the authors could show whether one of the members of the ensemble actually learns the correct causal model. Right now, there are some compelling results on MNIST, but it could be made clearer what the method is actually doing.”
>
> It is implicit in the results that one of the models in the collection does indeed learn to exploit the correct causal association. This is evidenced by the 'best' protocol results on the test sets in which the causally correct signal (i.e. the 'digit' signal) is by construction the only predictive signal present. We have not used the SEM example from the IRM paper as the presented benchmarks already demonstrate an ability to learn the causally correct predictive signal (in addition to any 'spurious' ones). We will add clarifying comments in a revision.
>
> > “The authors may want to clarify what “Linear” and “Best” refer to for a single model.”
>
> Thank you, we will clarify that under the 'linear' protocol, a new classifier is retrained on the output of the representation function (not the scalar output of the classifiers, in our notation) whereas for 'best' we simply choose the scalar prediction of the best performing frozen model (in this case of there is no choice to make given only a single model but we retrain a linear classifier under the 'linear' protocol).
>
> > (Clarification) “via reweighting of model outputs”
>
> To clarify, under the 'linear' protocol, we are training a linear model on the frozen, concatenated final layer representations, not the scalar predictions (in case you mean the latter when you say 'reweighting of model outputs').
>
> > (Random forests) “The authors may want to cite that literature, however.”
>
> A citation for random forests is included in the second paragraph of the related work. We agree that the motivation there is different i.e. averaging weak high-variance learners to improve performance of the aggregate vs our case of obtaining various strong learners each exploiting different solutions whose diversity is characterized by our proposed information theoretic measure.

---

### Author Response · Authors · 2020-11-23
**Revised paper submitted**

We have uploaded a revised version of the paper, making the following changes in light of reviewer feedback:
* Results for several additional baselines based on alternate diversity regularization terms
* Results for the unconditional version of our proposed estimator
* Further analysis of results, especially wrt explanations of behaviour of each model in a collection trained using our framework
* Further discussion on the motivation behind our evaluation protocol and the obstacles to considering alternate benchmarks and/or model selection strategies
* Additional citations and discussion to the related work section
* Clarification of any other points of misunderstanding raised by reviewers

---

### Decision · Program_Chairs · 2021-01-07
**Final Decision**

**Decision:**

Reject

**Comment:**

The paper tackles a major problem of supervised ML, that of the minimisation of the risk of a set of classifiers. This problem has received attention in numerous work over the past decades, much of which spans the formal aspects of the problem. The paper tackles the problem from a “diversity” standpoint. My main concern is, for such a problem and exhaustive formal and experimental SOTA, one cannot just evacuate any formal understanding of a contribution to future work (Authors’ reply to R2). The argument is then a victim of its own content, ending up in a sloppy vocabulary where “speculation” and “intuition” are called forward as justification to the calls for “rigorous” (R1) and “theoretical” understanding (R2 + answer to R2). I am confident the authors can find formal merit to their contribution, but this needs to be addressed. R1 + R4 hint on avenues to understand the contribution.